# Mechanisms of Plant Tolerance to RNA Viruses Induced by Plant-Growth-Promoting Microorganisms

**DOI:** 10.3390/plants8120575

**Published:** 2019-12-05

**Authors:** Igor V. Maksimov, Antonina V. Sorokan, Guzel F. Burkhanova, Svetlana V. Veselova, Valentin Yu. Alekseev, Mikhail Yu. Shein, Azamat M. Avalbaev, Prashant D. Dhaware, Gajanan T. Mehetre, Bhim Pratap Singh, Ramil M. Khairullin

**Affiliations:** 1Institute of Biochemistry and Genetics, Ufa Federal Research Center of the Russian Academy of Sciences, pr. Oktyabrya 71, 450054 Ufa, Russia; fourtyanns@googlemail.com (A.V.S.); guzel_mur@mail.ru (G.F.B.); veselova75@rambler.ru (S.V.V.); valentin-1994@yandex.ru (V.Y.A.); mikeshenoda@yandex.ru (M.Y.S.); avalbaev@yahoo.com (A.M.A.); krm62@mail.ru (R.M.K.); 2Department of Biotechnology, Aizawl, Mizoram University, Mizoram 796004, India; prashant.dhawre@gmail.com (P.D.D.); gtmehetre@gmail.com (G.T.M.); bhimpratap@gmail.com (B.P.S.)

**Keywords:** plant-growth promoting bacterium, plant viruses, RNase, plant defense, biocontrol

## Abstract

Plant viruses are globally responsible for the significant crop losses of economically important plants. All common approaches are not able to eradicate viral infection. Many non-conventional strategies are currently used to control viral infection, but unfortunately, they are not always effective. Therefore, it is necessary to search for efficient and eco-friendly measures to prevent viral diseases. Since the genomic material of 90% higher plant viruses consists of single-stranded RNA, the best way to target the viral genome is to use ribonucleases (RNase), which can be effective against any viral disease of plants. Here, we show the importance of the search for endophytes with protease and RNase activity combined with the capacity to prime antiviral plant defense responses for their protection against viruses. This review discusses the possible mechanisms used to suppress a viral attack as well as the use of local endophytic bacteria for antiviral control in crops.

## 1. Introduction

Viruses are obligate intracellular parasites that infect almost every living creature [1], including all cultivated crops. The majority of viruses that infect agricultural plants (at least 450 different species) are RNA viruses [2]. More than 40 of them infect potatoes causing cultivar damage which results in a significant loss in their productivity and deterioration of the tuber quality [3]. It is estimated that about 40% of total crop losses is caused by viral infection. DNA viruses are relatively rare in plants, compared to RNA viruses [4]. The effective viral disease management needs integration of all available strategies, which include avoidance of source of infection, vector control by various means, modification in cultivation practices, and the use of host resistance for viruses and vectors [5].

In the last two decades, RNA interference, a post-transcriptional gene-silencing approach, has been used to induce antiviral responses in plants with the help of genome-editing technologies, such as genetic transformation and the CRISPR/Cas9 (clustered regularly interspaced short palindromic repeats/Cas9 nuclease) system [6,7,8]. However, antisense mechanisms usually work for the RNA replicating inside the nucleus, like dsDNA-RT (Double-stranded DNA reverse transcriptase)viruses (caulimo- and badna-viruses) [8,9]. For plant protection against RNA viruses, ribozymes that are able to cleave viral RNA can be used [1,2].

Most of the viral diseases are spread by the vectors (insects [3,10], nematodes [3], pathogenic fungi [11,12], soil-born fungi [13], oomycetes [14], etc.), and therefore, they are a good target for preventing viral disease. One of the effective means to control virus vectors is the application of bactericides, insecticides, and fungicides.

Another approach is to develop virus-resistant (transgenic) plants capable of eliminating vectors by producing insect toxins, Cry and Vip proteins from plant-growth promoting microorganism (PGPM) *B. thuringiensis*, proteinase inhibitors, and antibiotics [9]. In contrast, currently there are no effective chemical means to eliminate viral particles in plants. Chemical antiviral drugs are being actively developed in India [15] and in the Netherlands [16]. The nucleosides’ analogs are potent antiviral agents, which effectively inhibit viral replication [17]. Besides the notable effects of these chemicals, they have some major drawbacks: expensiveness, extreme toxicity, and teratogenic effect on animals and humans [18].

Recently, signaling molecules which trigger the plant defense reactions against viruses have attracted the attention of researchers, since these molecules are less toxic, can be easily utilized by plants, and are destructed in the environment without accumulation of dangerous residues [19]. The application of chitin oligomers and PGPM contributed to a significant reduction of viral infection in plants [20,21,22]. The synthetic analogue of the salicylic acid (SA), benzo-(1,2,3)-thiadiazole-7-carbothioic S-methyl ester (benzothiadiazole), effectively reduced the spread of potato virus Y (PVY) in tomato plants [19,23]. It should be noted that the concentration of antiviral compounds in plants may decrease rapidly. Since viral particles are constantly present on the surface of plant cells, viruses may re-infect plants as soon as the inhibitory factor decreases [24].

Thus, the ability of PGPM to protect plants against pathogens and pests [25,26,27,28] makes them a significant resource for the development of biocontrol agents against plant viruses.

## 2. Microorganisms as the Means of Biocontrol of Plant Viral Infections

PGPM are beneficial microorganisms present in the rhizosphere that colonize plant roots. PGPM provide the host plant resistance to various biotic and abiotic stresses, in return the host plants provide them shelter and nutrients. Bacteria and fungi can live inside the plant tissues and these endophytic microorganisms form the closest relations with the host plant. In 1988, Clay [25] proposed that the plant pathogens may be inhibited by the endophytes, and the experimental confirmations were obtained by Siegel and Latch in 1991 [26].

Currently, several studies have confirmed that endophytic bacteria are highly effective against various pathogens and pests [27,28]. A systematic representation of different types of insecticidal metabolic products produced by endophytic strains of PGPM against different vectors is shown in Figure 1.

Interestingly, Bouizgarne [28] suggested that in some cases, the use of endophytes can be more beneficial than the generation of transgenic plants in terms of crop yield and virus protection. Modeling an artificial plant microbiome with PGPM that produce antiviral compounds and promote the defense potential of plants upon contact with viral particles can also become a promising alternative in the selection for virus tolerance [29].

The biocidal activity of rhizospheric and endophytic PGPM, which produce antibiotics (bacteriocins and/or lipopeptides), suggests that they display the indirect antiviral activity since phytopathogenic bacteria, protozoa, fungi, nematodes, and pests are vectors of a large number of viruses in plants (Figure 2).

Monitoring of world antiviral biocontrol agents market revealed no antiviral bio preparations in the classification of biopesticides or reports on direct acting of antiviral agents of biological origin [30]. The use of insecticidal and other biocidal microorganisms is a promising approach to control viral pest vectors. The agriculture industries are using bio preparations. For example, at the end of 2013, only in China, 132 bio preparations based on *Bacillus thuringiensis* were registered by 85 companies [31]. The biological preparation “Bitoxibacillin” (Sibbiopharm, Novosibirsk, Russian Federation) based on *B. thuringiensis* subsp. *thuringiensis* 98 (BtH_1_ 98) is well known in Russian Federation and the Commonwealth of Independent States [32]. But no data are available on the interactions of these biological agents with plants, and, as we suggest, a low efficiency of these agents observed in some cases arises from the lack of plant/PGPM relations resulting in a negative environmental influence on microorganisms.

The application of bio preparations based on bacteria with fungicidal and insecticidal activities against phytopathogens and insect pests should also help to reduce virus reproduction. Indeed, the treatment of sugar beet with the bacterium *Bacillus amylolequifaciens* drastically reduced the infection of sugar beet by *Phoma betae*, which is the vector of the Beet necrotic yellow vein virus (BNYVV) [12]. The application of *B. subtilis* BS3A25 strain has been found to reduce the CMV infection by inhibiting the development of its vector *Aphis gossipi* [33]. Colonization of the internal tissues of onion plants with the endophytic fungus *Hypocrea lixii* (F3ST1) reliably reduced the replication of IYSV as well as the feeding activity of its main vector, *Thrips tabaci* Lindeman [34]. 

A large number of studies documented the activity of PGPM against viral infection, virus spread, and reproduction in plants (Table 1). Unfortunately, no reports are available on the interactions of these bacteria with host plants, including their possible penetration into the internal tissues of plants.

Thus, the soil drench, seed or root treatment, and foliar spraying with PGPM (probably, endophytic, no data) microorganisms contributed to a lesser degree of viral disease symptoms in plants as well as to a reduction of virus concentration (Table 1). This information is very important for PGPM’s practical application. Thus, soil treatment with *P. putida* A3 prior to sowing reduced TMV infection in tobacco plants more effectively than soil treatment with this PGPM after sowing [51]. The effectiveness of seed treatment is demonstrated in many cases (Table 1).

It should be noted that some strains are able to protect a broad spectrum of plant species, enabling the development of versatile biocontrol agents on the base of these strains. Thus, treatment of plants with *P. fluorescens* strain CHA0 induced resistance against TNV in tobacco [44] as well as against ULCV in black gram (*Vigna mungo*) [45] and *Musa* sp. [20]. Cucumber and tomato seed treatment by *Serattia marcescens* 90-166 strain in combination with *P. putida* 89B-61 strain [58] or *B. pumilus* SE34 strain [59] respectively, induced resistance against CMV, markedly reducing the disease symptoms.

Importantly, combinations of several strains can be more effective against viruses than individual strains. Thus, the dual PGPM combinations, each including strain *Bacillus subtilis* GB03 and one of the following strains: SE34 (*B. pumilus*), IN937a (*B. amyloliquefaciens*), IN937b (*B. subtilis*), INR7 (*B. pumilus*), or T4 (*B. pumilus*), effectively protected tomato plants against CMV [57]. Plant growth-promoting microbial consortium, including *Bacillus licheniformis* MML2501 *+ Bacillus sp.* MML2551 *+ Pseudomonas aeruginosa* MML2212 *+ Streptomyces fradiae* MML1042, reduced the damage to sunflower plants caused by sunflower necrosis virus disease (SNVD), much stronger than did individually tested strains [60]. Moreover, the addition of *Streptomyces* sp. PM5 and *Trichothecium roseum* MML005 to this microbial consortium enhanced its defense effect [60]. Treatment of papaya and tomato seeds with PGPM mixture, consisting of *B. amyloliquefaciens* IN937a, *B. pumilus* SE34, and *B. pumilus* T4, contributed to the subsequent protection of papaya and tomato plants from Papaya ringspot virus (PRSV-W) and tomato chlorotic spot virus (TCSV), respectively [61].

Individual strains can be used in combination with ecologically friendly compounds, which are effective against plant diseases, such as well-studied chitin and chitosan. The combination of chitin oligomers and *Pseudomonas fluorescens* CHA0 allowed for inducing systemic resistance against BBTV in banana plants [20]. Similarly, seed treatment with PGPM solution containing *Bacillus polymixa* and *Pseudomonas fluorescens* mixed with chitosan reduced SqMV infection in cucumber plants [22], while the application of *Pseudomonas* sp. (206 (4) + B-15 + JK-16) in combination with chitosan enhanced the protection of tomato plants against ToLCV [21].

We suggest that the ability of PGPM to act jointly in consortiums is an opportunity for the development of effective and diversified microorganisms containing antiviral products for plant protection.

## 3. Bacterial Nucleases and Their Antiviral Activity

Bacteria can directly bind and destroy viral particles, by secreting extracellular proteases, nucleases, and proteases. Thus, *P. putida* A3 [51] and *B. pumilus* [52] were shown to destroy virus particles in the juice from tobacco leaves infected with TMV. In this regard, an alternative strategy for protecting plants from viruses can be based on the use of microbial enzymes, for example, extracellular nucleases. Currently, more than 20 extracellular RNases of Bacillus have been discovered. For example, *B. amyloliquefaciens*, *B. pumilus*, and *B. licheniformis* produce extracellular RNases called barnases, binases, and baliphases, respectively [63,64]. Among endophytic microorganisms isolated from diverse cultivated *Cucurbitaceae*, 73% of *Bacillus*, 27% of *Paenibacillus,* and 30% of *Enterobacteriaceae* isolates, as well as all *Cronobacter*, *Pantoea*, *Microbacterium*, and *Staphylococcus* isolates displayed RNase activity [65]. Interestingly, the secreted RNase (Bsn, 241 amino acids) from *B. subtilis* encoded by the *bsn* gene [66] is closely related in structure and in its enzymatic properties to the *B. pumilus* nuclease binase II (292 amino acids) encoded by the *birB* gene [67].

It has long been known that the treatment of plants with RNases reduces the virus accumulation. A high correlation was shown between the RNase activity and virus resistance in different potato varieties [68]. Recent studies have demonstrated that bacterial RNases effectively inactivate RNA-containing viruses in plants by cleaving their RNA and disrupting the formation of a virus coat [69]. It has been established that *B. cereus* ZH14 produces a new type of extracellular RNase which is active against TMV [70]. *B. pumilus* RNase directly suppressed the development of potato virus S (PVS) and potato virus M (PVM) infection and also reduced the red clover mottle virus (RCMV) incidence in pea plants [69]. Moreover, treatment of tobacco plants with 100 µg/mL RNase almost completely inhibited PVX infection (94%) [69].

In addition to developing increased resistance against viral infection, bacterial barnases can participate in plant protection against other diseases. For example, transgenic tobacco plants producing barnase were shown to be protected from the late blight disease [71]. It has been found that potato plants expressing *S. marcescens* nuclease display enhanced resistance to pathogens [71,72]. Thus, genetic transformation using a bacterial RNase gene may be a promising approach for the engineering of plants with resistance to viral infection [73,74]. Soybean plants expressing the *Schizosaccharomyces pombe* PAC1 RNase gene display resistance to a wide range of viruses [75]. A genetically engineered CRISPR/Cas13a construct containing class 2 type VI-A RNase capable to recognize and cleave single-stranded RNA was introduced into the *Nicotiana benthamiana* genome and effectively reduced the Turnip mosaic virus (TuMV) incidence [76]. Approximately one-third of the transgenic tobacco plants expressing the *B. amyloliquefaciens* barnase gene was found to be fully resistant to TLCV infection [77]. Thus, the selection of endophytic microorganisms that can produce RNases directly in plant tissues is a promising strategy for the development of virus control mechanisms in plants.

## 4. Signal Pathways and Mechanisms of Plant Resistance to Viruses Induced by Microorganisms

### 4.1. Virus Recognition and Systemic Resistance in Plants

The plants themselves have quite effective defense mechanisms that prevent viral infection and virus spread. For instance, there are two main types of virus resistance in *Solanaceae*: extreme resistance and localized hypersensitivity [3]. Extreme resistance provides high resistance to all strains of the virus while localized hypersensitivity is strain-specific. It is necessary to note that *Solanum tuberosum*, the most important cultivated *Solanaceae*, have no defense genes against the most harmful PVX and PVY.

Hypersensitive response (HR) is characterized by necrosis and disruption of the virus systemic spread in plants. In potato plants, HR in response to strains PVYC and PVYO of PVY is controlled by potato *Nytbr* and *Nctbr* genes, respectively [3,78].

It was shown that the avirulence factor of the PVY virus is the helper component proteinase (HC-Pro) cistron of PVY, while *Nx*-mediated hypersensitivity and *Rx*-mediated extreme resistance were elicited by different subunits of coat protein (CP) of PVX [79]. CP of a virus as well as viral RNA, the so-called pathogen-associated molecular patterns (PAMPs), are recognized by plant cell receptors, leading to the development of defense responses in plants [80], including rapid generation of reactive oxygen species (ROS), changes in the content of phytohormones, synthesis of other metabolites, as well as the induction of local and systemic expression of defense genes [81]. This mechanism does not appear to be associated with the RNA interference, which is also an important strategy to protect plants against RNA-containing viruses [82].

When analyzing the protective effect against viral infection, it should be considered that the substances produced by PGPM may interact with the host’s immune system, thereby inducing specific responses in plants. For instance, endophytic bacteria themselves induce the systemic resistance in plants against pathogens [83,84], i.e., they trigger plant defense responses as weak pathogens [85]. This is likely due to the fact that various PAMPs, including flagellin and lipopeptides of endophytic bacteria [39] or CP of viruses [86,87], are recognized by receptors containing leucine-rich repeats (LRR) [88]. Genes encoding pathogenesis related (PR) proteins PR-4 and PR-10 with antiviral activity, including RNase activity, are known to be expressed in plants under the influence of rhizobacteria and their metabolites [56,89], as well as in response to viral [89] and fungal [90] infections. Thus, PGPM can prime plant reactions to viral infection.

### 4.2. Plant-Growth Promoting Microorganism (PGPM) and Regulation of Plant Defense Mechanisms Against Viruses

Pro-/antioxidant enzymes and phenolpropanoid metabolism enzymes were shown to be involved in plant defence reactions induced by PGRB and their metabolites. Thus, treatment of banana plants by the mixture of rhizospheric *P. fluorescens* Pf1 and endophytic *Bacillus* spp. EPB22 resulted in the activation of peroxidase, polyphenol oxidase, and phenylalanine ammonia lyase (PAL), as well as the accumulation of phenolic compounds, which contributes to multiple decreases in BBTV incidence with a final efficiency of up to 80% [62]. Similar changes were observed in BBTV-infected banana [20], ULCV-infected black gram [45], and TSWV-infected tomato plants [91].

According to modern concepts, the plant defense response against pathogens and pests with different lifestyles is regulated by the balance of jasmonic acid (JA)- and salicylic acid (SA)-mediated signaling pathways. In most of the studies, PGPM are characterized as microorganisms that activate resistance to a wide variety of herbivores and necrotrophic pathogens by the JA-dependent signaling pathway, leading to the development of induced systemic resistance (ISR). At the same time, SA induces biochemical processes in plants, leading to resistance against biotrophic pathogens, viruses [1,3], and pests (hemiptera and aphids) [92]. The ability of the *B. subtilis* BS3A25 isolate to reduce the melon aphid *Aphis gossipi* (CMV vector) population [33] may be due to aphicidal activity of bacteria-produced surfactants [92,93]. Surfactin of *B. subtilis* BMG02 increased the resistance of tomato plants to tomato mosaic virus (ToMV) by triggering rapid H_2_O_2_ generation and the expression of salicylate-sensitive genes encoding PR-2 protein and PAL, the latter participating in SA biosynthesis [94].

Maurhofer et al. [46] showed that the ability of *P. fluorescens* CHA0 to protect tobacco plants from TMV is associated with the systemic accumulation of SA in plants as well as with the accumulation of PR-1a, PR-1b, and PR-1c proteins. It can be assumed that the ability of *Pseudomonas spp.* to protect plants against viral infection is due to SA-induced systemic resistance and is associated with the local generation of ROS in the infection zone. However, the use of bacterial mutants with disrupted production of SA and pseudobactin allowed to show that production of these metabolites by *P. fluorescens* WCS374r are not required for eliciting Induced systemic resistance (ISR) in Arabidopsis against *P. siringae* [95], suggesting a fundamental role of JA in this process.

*Bacillus* spp. associated with tobacco plants induced the development of systemic resistance against TMV by inhibiting the synthesis of CP and enhancing the expression of genes encoding JA- and SA-signaling pathways proteins, Coil and NPR1, defense proteins PR-1a and PR-1b, and cell-wall expansins NtEXP2 and NtEXP6 [39]. The *Rhodopseudomonas palustris* GJ-22 strain, capable of producing Indolil-acetic acid and 5-aminolevulinic acid, reduced TMV incidence in tobacco plants in the field conditions. Genes of both salicylate-(*NbPR1a* and *NbPR5*) and jasmonate-mediated (*NbPR3* and *NbPDF1.2*) signaling pathways were activated after the treatment with this strain [54]. This is somewhat contrary to the data showing that the treatment of tomato with the *B. amyloliquefaciens* MBI600 strain induced plant resistance to TSWV and PVY accompanied by gene expression of only the SA-induced signaling pathway [24].

Meanwhile, Ryu et al. [55] have revealed that the application of the *S. marcescens* 90-166 strain to *Arabidopsis* plants induced resistance against CMV independently of SA, but dependent on JA. It was revealed that the regulatory activity of *B. amyloliquefaciens* FZB42 [96] and *B. cereus* AR156 [97] is associated with their ability to inhibit the mechanism of RNA interference of the suppressor genes of the JA defence pathway involving micro RNAs, miR846 and miR825/miR825*, respectively. Nazari et al. [98] revealed the upregulation of miRNAs, nta-miR167 and nta-miR393, and the accumulation of flavanoid compounds in tobacco plants inoculated with *B. subtilis* ATCC21332 and subsequently infected by *Agrobacterium tumefaciens* IBRCM10701. The authors suggested that the expression of these miRNAs as well as the accumulation of flavonoid derivatives may be used as markers to assess the efficiency of the PGPM defense effect [98].

The treatment of pepper plants with *B. amyloliquefaciens* 5B6 reduced the CMV incidence in the field conditions [56] associated with the induction of transcription of genes encoding PR-4, PR-5, and PR-10 proteins. For instance, in hot pepper *Capsicum annuum,* bacterial derived 2,3-butanediol has been shown to develop a defense response to CMV and TMV, evidenced by the accumulation of transcripts of various defense marker genes, such as *Capsicum annuum pathogenesis-related 4* (*CaPR4*), *Ca chitinase 2* (*CaChi2*), *Ca phenylalanine-I ammonia-lyase* (*CaPAL*), *CaSAR8.2*, *Ca 1-aminocyclopropane-1-carboxylic acid oxidase* (*CaACC*), and *Ca proteinase inhibitor 2* (*CaPIN2*), which was similar to the increase in expression of those genes in the benzothiadiazole-treated plants [99].

Beris et al. [24] explained the development of PGRB-induced tomato resistance to the spotted wilt virus and PVY by simultaneous expression of defense genes predominantly of SA-, and to a lesser degree, of JA-signaling cascades. Thus, it should be noted that the treatment of plants with benzothiadiazole, which is used as a reference in many studies of plant resistance to viruses, despite a decrease in the incidence of viral particles in plants, in some cases, suppressed growth and reduced the weight of plants during viral infection compared with bacterial cultures [54,55]. As it was shown by Kumar and co-workers [49], soil application of *Paenibacillus lentimorbus* B-30488 enhanced the resistance of tobacco plants to CMV while it maintained photosynthetic activity and plant growth. At the same time, the activity of antioxidant enzymes decreased, expression of genes encoding pathogen-induced proteins increased, and polyphenols accumulated, which subsequently prevented the virus spread through plant tissues [49]. Consequently, continued research is needed to develop new approaches to enhance the efficiency of PGPM for improving plant immune potential.

## 5. Endophytic PGPM as Vectors of RNA Insecticides: Future Approaches

RNA interference (RNAi), which acts at the transcriptional level through RNA-directed DNA methylation and at the post-transcriptional level is mediated by Dicer-like RNase III and small interfering RNA (siRNA), which recognize and inactivate viral RNA, may be used to obtain viral-resistant plants [100]. Zhan and colleagues [101] showed that potato lines, which express CRISPR/Cas13a constructs containing small guide RNA (sgRNA) against coding regions of PVY, were distinguished by lower amounts of virus particles in the tissues and, respectively, by lowered disease symptoms.

Besides the potent effectiveness against virus infection, for example, of tomatoes against the DNA-containing tomato yellow leaf curl virus (TYLCV), expression of the RNAi transgenes affected the host plant transcriptome, resulting in slight phenotypic and developmental abnormalities of the transgenic plants [102]. The bio-insecticides which work on RNA interference are the best way to control the plant virus spreading pest [103,104]. In 2016, Whitten et al. [105] considered that it is necessary to use RNA interference for plant protection against insects, viruses, and fungal phytopathogens. However, it is too tedious and impractical to design such an oversize construct (comprising the defense genes against insects, viruses, and fungi), to integrate it into the plant, preserve it in the plant cells, achieve its biosafety, and subsequently develop the “super resistance” in the target plants. Interestingly, there are new means to deliver “RNA insecticides” in pests’ organisms for their elimination with the help of highly specific microsymbionts exclusively from certain insect species [106]. Monsanto announced the launch of a first insecticide based on RNA silencing technology in the 2020s [107]. This scheme is fundamentally different from plant protection with the use of genetic modifications; however, it requires methods to transfer “RNA insecticide”, “RNA fungicide”, or “RNA viricide” into the plant as well as to protect RNA molecules from sunlight and rain washout [106]. It is likely that endophytic bacteria may also be used as delivery vectors known to successfully colonize plant tissues and form protective biofilms in the apoplast.

## 6. Conclusions

Currently, a large body of data has accumulated on the positive effect of bacteria and their metabolites on the enhancement of plant defense against viral infection. At the same time, it should be noted that there are different mechanisms of plant defense against viral infection induced by rhizospheric, endophytic, and symbiotic bacteria, as well as their metabolites. PGPM-induced plant defense against common pests, fungal, and bacterial pathogens [85] play an important role in preventing the transmission of viral infections. Several studies have suggested the possibility of using bacterial as well as plant RNase to protect plants from pathogenic viruses. Thus, the identification of the biological properties and the role of PGPM (in particular, endophytic strains) in plant microbiome with the aim of developing biological products with comprehensive activities (antiviral, insecticidal, fungicidal, bactericidal, immune, and growth-promoting), which will be environmentally safe products for plant protection against diseases and pests, is a promising approach of plant defense against viruses (Figure 3). In order to develop a complex multifunctional biological product of a triple action (insecticide + fungicide + viricide), it is important to investigate the plant signaling pathways induced after the influence of these preparations. Thus, it is necessary to study the crosstalk between signaling pathways involved in the development of resistance induced by endophytic bacteria with different biological activities that will eventually contribute to the development of preparations against a wide range of pathogens and herbivores.

PGPM can influence the virus spread by direct antiviral effects of RNase-producing microorganisms or systemic resistance-inducing microorganisms, which live on the surfaces and/or in the internal tissues of plants. These microorganisms can indirectly decrease viral load in agroecosystems by the control of vectors, in particular, by “RNA biocides” specific for pests. PGPM can be “useful” in various combinations for the development of biocontrol agents that will combine direct and indirect activities with close relations with host plants.

## Figures and Tables

**Figure 1 plants-08-00575-f001:**
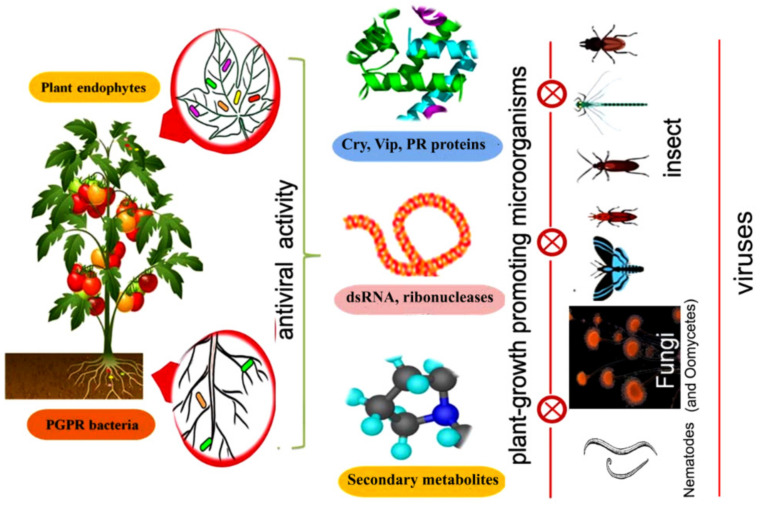
Effectiveness of different biocidal metabolites produced by plant-growth promoting microorganism (PGPM) for the control of insect vectors and their associated phytopathogens.

**Figure 2 plants-08-00575-f002:**
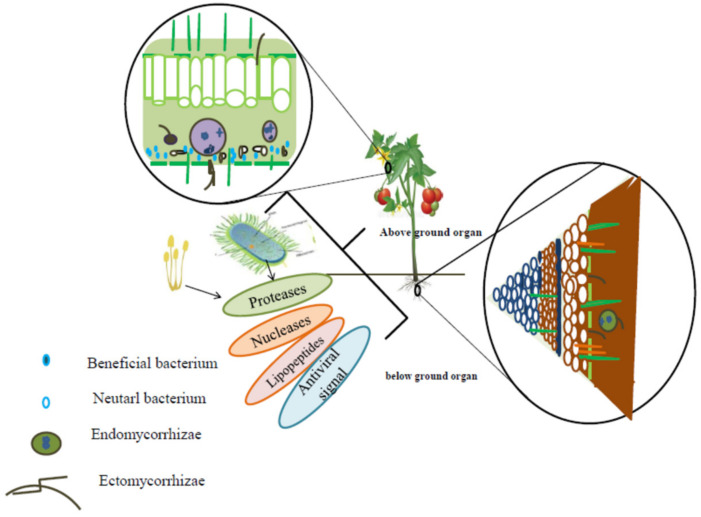
Location of PGPM in plant tissues and their secondary metabolites, which are of benefit for plants protection against biotic stressors.

**Figure 3 plants-08-00575-f003:**
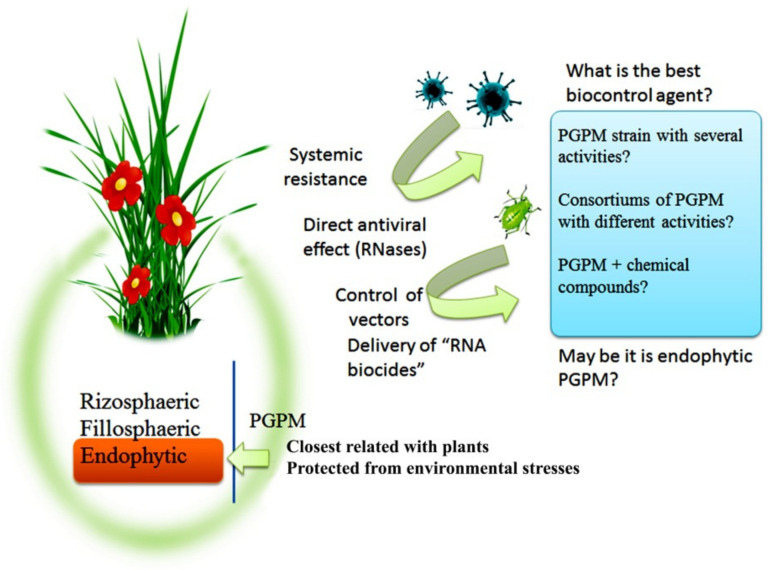
Effects of PGPM that may promote the development of plant protection to viral diseases and prospects of PGPM use.

**Table 1 plants-08-00575-t001:** Protection of plants by plant growth-promoting microorganisms (PGPM) against viral diseases.

Strain of PGPM	Plants	Method of Treatment	PGPM Effectiveness Against Viruses	Source
*Pseudomonas syringae* (heat-killed cells)	*Nicotiana tabacum*	Foliar spraying	Tobacco mosaic virus (TMV)	[35]
*Bacillus uniflagellatus*	*Nicotiana tabacum*	Soil drench	TMV	[36]
*Pseudomonas lachrymans*	*Cucumis sativus*	Foliar spraying	Cucumber mosaic virus (CMV)	[37]
*Bacillus* spp.	*Gossypium herbaceum*	Soil drench/foliar spray	Tobacco streak virus (TSV)	[38]
*B. amyloliquefaciens* MBI600	*Solanum lycopersicum/Solanum tuberosum*	Foliar spraying	Tomato spotted wilt virus (TSWV)	[24]
*B. amyloliquefaciens* FZB24, FZB42	*Nicotiana tabacum*	Soaking the roots	TMV	[39]
*B. pumilus* EN16	*Nicotiana tabacum*	Soaking the roots	TMV	[40]
*B. subtilis SW1*	*Nicotiana tabacum*	Soaking the roots	TMV	[40]
*B. subtilis* Ch13	*Solanum tuberosum*	Micro-tubers spraying	PVY, potato virus X (PVX)	[41]
*Bacillus pumilus* T4	*Vigna unguiculata*	Seed spraying	Bean Common Mosaic Virus (BCMV)	[42]
*Bacillus subtilis* GBO3	*Vigna unguiculata*	Seed spraying	BCMV	[42]
*B. pumilus* SE34	*Solanum lycopersicum*	Seed spraying	CMV	[43]
*P. fluorescens* CHA0	*Nicotiana tabacum*	Foliar spraying	Tobacco necrosis virus (TNV)	[44]
*P. fluorescens* CHA0	*Vigna mungo*	Foliar spraying	Urdbean leaf crinkle virus (ULCV)	[45]
*P. fluorescens P3*	*Nicotiana tabacum*	Foliar spraying	TNV	[46]
*Bacillus cereus* (I-35), *Stenotrophomonas* sp. (II-10)	*Capsicum annuum*	Seed treatment and soil drench	TMV virus, Chili veinal mottle virus (ChiVMV)	[47]
*Pseudozyma churashimaensis*	*Capsicum annuum*	Soil drench	CMV, pepper mottle virus (PMV), pepper mild mottle virus (PMMV), and Broad bean wilt virus (BBWV)	[48]
*Paenibacillus lentimorbus B-30488*	*Nicotiana tabacum*	Soil drench	CMV	[49]
*Azotobacter vinelandii*, *Azotobacter chroococcum*	*Solanum tuberosum*	Tuber drench	PVY, PVX, PLRV	[50]
*P. putida* A3	*Nicotiana tabacum*	Soil drench	TMV	[51]
*B. pumilus*	*Nicotiana tabacum*	Leaves juice	TMV (destroying viral particles)	[52]
*Bacillus pumilus* SE34, *Kluyvera cryocrescens* IN114, *Bacillus amyloliquefaciens* IN937a, *Bacillus subtilus* IN937b	*Solanum lycopersicum*	Soil drench	CMV	[53]
*Rhodopseudomonas palustris GJ-22*	*Nicotiana tabacum*	Seed treatment	TMV	[54]
*S. marcescens 90-166*	*Arabidopsis thaliana*	Seed treatment	CMV	[55]
*B. amyloliquefaciens 5B6*	*Capsicum annuum*	Foliar treatment	CMV	[56]
Microbial consortiums				
*Bacillus subtilis* GB03 + *B. pumilus* SE34/*B. amyloliquefaciens* IN937a/*B. subtilis* IN937b/*B. pumilus* INR7/*B. pumilus* T4	*Solanum lycopersicum*	Seed treatment	CMV	[57]
*Serattia marcescens* 90-166 + *P. putida* 89B-61/*B. pumilus* SE34	*Cucumis sativus*	Seed spraying	CMV	[58,59]
*Bacillus licheniformis* MML2501 + *Bacillus* sp. MML2551 + *Pseudomonas aeruginosa* MML2212 + *Streptomyces fradiae* MML1042	*Helianthus annuus*	Seed treatment	Sunflower necrosis virus disease (SNVD)	[60]
*B. amyloliquefaciens* IN937a + *B. pumilus* SE34 + *B. pumilus* T4	*Carica papaya /Solanum lycopersicum*	Seed treatment	Papaya ringspot virus (PRSV-W) /Tomato chlorotic spot virus (TCSV)	[61]
*P. fluorescens Pf1. + Bacillus spp. EPB22 incidence with a final efficiency of up to 80%*	*Musa spp*	Foliar treatment	Banana bunchy top virus (BBTV)	[62]
Individual strain/microbial consortiums + chemical immunoregulators				
*P. fluorescens* CHAO+chitin	*Musa* spp.	Soil drench	BBTV	[20]
*Pseudomonas* sp. (206 (4) + B-15 + JK-16 + chitosan olygomers	*Solanum lycopersicum*	Seed treatment	Tomato leaf curl virus (ToLCV)	[21]
*Bacillus polymixa* + *Pseudomonas fluorescens* + chitin olygomers	*Cucumis sativus*	Seed treatment	Squash mosaic virus (SqMV)	[22]
*B. pumulus* INR7 + benzothiadiazole	*Capsicum annuum*	Foliar treatment	CMV	[23]

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
