# Peer review of "Mechanisms of Plant Tolerance to RNA Viruses Induced by Plant-Growth-Promoting Microorganisms"

_plants, 2019, doi:10.3390/plants8120575_

Round 1
Reviewer 1 Report
The Ms by Maximov et al entitled “Mechanisms of plant tolerance to viruses induced by plant-growth-promoting microorganisms” deals with a very interesting subject that could surely fit within the scope of the journal.
The authors start from the point that conventional methods for reducing the impact of viruses in agriculture are not adequate and nonconventional methods are an alternative (the sentence reads “As the conventional viral disease control is not efficient the nonconventional methods being employed”). Beside the fact that this starting sentence is not correct from the syntax point of view, there is confusion in what is Conventional and Modern (see Chapter 2) and what is innovative and nonconventional.
The content of the second Chapter 2. Modern approaches to control viral infections in plants describes a list of possible strategies used to reduce viral infections in plants, but some of them are not modern at all. Moreover, the list is not well organized, with a mix up of different sentences, sometimes with no clear rationale. For example, in the 2.3 Control of vectors paragraph, the authors say (line 74) “The transgenic plants are not 100% protected from any virus, for example, transgenic potato expressing viral CP showed almost 100% resistance to infection by potato virus Y (PVY) but variable resistance (72 - 96%) to the infection by potato leafroll virus (PLRV) [10]”, a sentence that is “nonsense” in this specific context.
The same is true for other sentences in the same paragraph, such as (lines 64) “Thus, the aphids are a seasonal creature the virus disease can be avoided by simply avoiding the season of vectors. The next strategies are to target the virus vectors directly as it is not simply practical for changing the planting season”, or (line 93) “Bactericides, fungicides, and insecticides should be used for protecting plants during the development as phytopathogens and insect pests are potential vectors of viruses.”, Even when correct, the meaning of these sentences here is not clear.
The true object of the review starts with Chapter 3. Microorganisms as a means of biocontrol of plant viral infections (lines 124). The authors provide an interesting list of cases where bacteria used as PGPR were beneficial as they interfered somehow with viral symptoms or viral infection. However, there is some confusion in listing the different examples: I suggest to re-organize the Chapter, for example dividing the use of bacteria applied to leaves or aerial parts of the plant from bacteria applied to roots. Moreover, they could distinguish the use of live bacteria from killed ones or from the treatments with chemical substances produced by bacteria.
Also in this chapter, there are numerous sentences that should be reorganized, such as for example “Plant growth-promoting rhizobacteria (PGPR) are known to exert beneficial effects on vegetatively propagated crops, such as increased growth and enhanced resistance to abiotic and biotic environmental factors, including phytovirus infection [44].” This sentence is placed almost at the end of the chapter dealing with PGPR, so not in the correct place.
The role of the Chapter 7. Generation of transgenic plants is not clear. The generation of transgenic plants is a technology, not specifically related per se to the object of this Review. If the authors intend to refer to the use of dsRNA-based constructs or hairpin constructs, or, more generally to the HIGS and SIGS strategies, this is not clearly mentioned. Even in this case, what is the connection with the title of the review?
In sum, even if many interesting examples are reported that warrant consideration by plant scientists, the Ms is immature and requires a careful re-organization, with a more logical description.
There are also several misspelled words and/or wrong sentences all along the Ms that do not help to critically and constructively evaluate it. Accurate editing by native English speaking people is necessary. Special attention should be given also to scientific names in latin and to virus names. Virus naming must follow the recent ICTV guidelines, avoiding repetitions of full virus names especially when they are cited several times (see for example cucumber mosaic virus that is mentioned many times, every time misspelled or named in a wrong way). Once cited, the virus acronym can be used.
Reviewer 2 Report
The manuscript entitled "Mechanisms of plant tolerance to viruses induced by plant growth promoting microorganisms" by Maksimov et al, is reviewing the possibility of microorganisms that interact with plants or with viral vectors to protect agains the spread or replication of the viruses.
The authors base their manuscript in the premise that most plant-infecting viruses are RNA based (which is true) but fail to address why not using the same methodology to attempt to find a way to stop the spread of DNA plant viruses.
The manuscript at this point seems disorganized to this reviewer and has a serious issue with run on sentences and some sentences that doesn't seem to make sense the way they are built.
Furthermore, two very nice figures are presented that are related to the introductory paragraphs but the bulk of the manuscript is very dry and the addition of another figure or two that explains the main ideas for the manuscript will be very welcome.
The authors are repetitive with several ideas through the manuscript and also seem to abuse referencing work in several places, where they display five to ten examples of what is described but only as a list of authors and their main finding. This makes the manuscript difficult to read, it can be easily placed in a Table that can be referred to if needed.
The proposal to change the microbiome of plants is confused sometimes with changing endophytes so it is not clear what is the main strategy proposed by the authors.
In the conclusions, the authors actually make a nice summary of what they propose in the text, if the actual text was organized as linearly as the conclusion, it will help a lot to follow up the author’s ideas and proposals.
Some specific comments regarding the manuscript are listed below:
In the introduction, lane 31. The sentence is difficult to read “As the conventional….”
Also, it would be useful to have Sanford and Johnston citation.
Lane 48 “cucumber plant”, should read cucumber plants.
Lane 56 “non-enzymatic reactions”, since ribozymes are considered enzymes, the phrase may be …”RNA that can perform reactions in the absence of proteins,”
Lane 64: This sentence is not clear, it can be, as example: “The aphids are seasonal therefore the viral transmission carried by it can be avoided by planting during the aphid off-season”.
Lane 62: Maybe better “Vector control”?
Lane 74: It seems like a title has been lost here since the whole paragraph deals with transgenics, then pesticides, then chemical antivirals, etc. Maybe the title “Control of vectors” can be changed or separated in smaller chunks.
Lane 93: This paragraph is too long and the point can be made much earlier to facilitate reading.
Lane 115: The species for Capsicum is truncated and probably corrected by auto-correction features (replace annual by annuum)
Lane 184: This paragraph is also lost in the number of citations and a big list of facts, with little connection to an idea or proposal.
Lane 220: Another list of facts that can be reduced to a couple of paragraphs with more elegant description.
This reviewer thinks that the manuscript will greatly benefit from a figure describing the ideas placed in the paragrpahs starting with Lane 275. “Signal pathways and mechanisms….”
Lane 397: TYLCV is a DNA virus but is not stated in the text, it could be useful to mention it.
Lane 404: Why is Monsanto placed in quotes?
Lane 396: The ideas proposed in this paragraph are interesting, but it seemed diluted, maybe make a section for them? Like: future approaches?
Lane 422: The authors named three possible bacterial categories that can be used in the text to better discuss their strategies and proposals. It seems good to have rhizospheric separated from endophytic and symbiotic and separate their usefulness.
As a final comment: the ideas and proposal of using microbes on a more thorough mode to control viruses and viral vectors seem very interesting, however the organization and writing can be greatly improved.
Round 2
Reviewer 1 Report
In this revised version the authors have just made some small improvement, but the major issues raised at the first revision have not been solved.
The rational and organization of the Ms are not yet sufficiently clear and the reader is lost after too many repetitive concepts. Chapter 2 is out of the scope of the title of the Ms and, since it is not well organized, it should be eliminated "tout court". It could be substituted by an introductory sentence for the use of microorganism as Plant growth promoting bacteria to control viruses, indirectly (for their activity against virus vectors, such as insects, fungi, nematodes) or directly (for the production of antiviral substances, antiviral enzymes, modulation of plant response against pathogens, via hormones or through other pathways).
Unfortunately, in spite of the interest in this topic, I cannot recommend this Ms for publication.
Author Response
We thank reviewer for his/her special attention to this matter. We do appreciate it. We regret that our manuscript does not meet the requirements.
Reviewer 2 Report
The authors claim that “no serious attempts have been made for the control of DNA viruses”, which is a false statement, just see Rojas et al, 2018 (Annu. Rev. Phytopathol. 2018. 56:637–77).
In lane 88 the statement starts with “The”, it should be “One”, since it is not the only way to control vectors.
Lane 114, PGPM is not defined, please do and remove the definition from lane 127.
In lane 139 can be stated that “for some cases”, instead of such a general claim for the use of endophytes instead of transgenic technologies.
Table 1 is not properly named, it has the “1” missing.
Also, in the table it can be seen that Solanym lycopersicumd has some accents added to the a and e of each of the names, in all the cases. Can they be removed? They are also present in Carica papaya.
Lane 266, there is an extra “e” in Solanaceae.
A longer description of for Figure 3 (Figure legend) will enhance the understanding of the proposed mechanisms.
Author Response
We thank reviewer for his/her special attention to this matter.
With regard to reviewer issues we would like to make the following replies:
The authors claim that “no serious attempts have been made for the control of DNA viruses”, which is a false statement, just see Rojas et al, 2018 (Annu. Rev. Phytopathol. 2018. 56:637–77).
We are afraid that it was a misspelling. Now:
Serious attempts have been made for the management of DNA viruses. Conventional control measures (chemical, cultural and physical) against the vectors are the most widely used under the field conditions [4].
It is not contrary to the Rojas et al, 2018, since this work is devoted to these measures.
In lane 88 the statement starts with “The”, it should be “One”, since it is not the only way to control vectors.
It was replaced.
Lane 114, PGPM is not defined, please do and remove the definition from lane 127.
It was defined above.
In lane 139 can be stated that “for some cases”, instead of such a general claim for the use of endophytes instead of transgenic technologies.
It was specified.
Table 1 is not properly named, it has the “1” missing.
It was corrected.
Also, in the table it can be seen that Solanym lycopersicumd has some accents added to the a and e of each of the names, in all the cases. Can they be removed? They are also present in Carica papaya.
It was corrected.
Lane 266, there is an extra “e” in Solanaceae.
It was corrected.
A longer description of for Figure 3 (Figure legend) will enhance the understanding of the proposed mechanisms.
It was descripted:
Fig. 3. Effects of PGPM that may promote the development of plant protection to viral diseases and prospects of PGPM use.
PGPM can influence the virus spread by direct antiviral effects of RNase-producing microorganisms or systemic resistance-inducing microorganisms, which live on the surfaces and/or in the internal tissues of plants. These microorganisms can indirectly decrease viral load in agroecosystems by the control of vectors, in particular, by“RNA biocides” specific for pests. PGPM can be “useful” in various combinations for the development of biocontrol agents that will combine direct and indirect activities with close relations with host plants.
